# Subcritical Water Extraction of *Undaria pinnatifida*: Comparative Study of the Chemical Properties and Biological Activities across Different Parts

**DOI:** 10.3390/md22080344

**Published:** 2024-07-27

**Authors:** Jin-Seok Park, Ji-Min Han, Sin-Won Park, Jang-Woo Kim, Min-Seo Choi, Sang-Min Lee, Monjurul Haq, Wei Zhang, Byung-Soo Chun

**Affiliations:** 1Department of Food Science and Technology, Pukyong National University, 45 Yongso-Ro, Nam-Gu, Busan 48513, Republic of Korea; jin1931@pukyong.ac.kr (J.-S.P.); wlals383@gmail.com (J.-M.H.); psw3475@gmail.com (S.-W.P.); corn6746@naver.com (J.-W.K.); choiminseou@naver.com (M.-S.C.); 2sangmin0630@naver.com (S.-M.L.); 2Institute of Food Science, Pukyong National University, 45 Yongso-Ro, Nam-Gu, Busan 48513, Republic of Korea; mr.haq@just.edu.bd; 3Department of Fisheries and Marine Bioscience, Jashore University of Science and Technology, Jashore 7408, Bangladesh; 4Centre for Marine Bioproducts Development, College of Medicine and Public Health, Flinders University, Adelaide, SA 5042, Australia; wei.zhang@flinders.edu.au

**Keywords:** *Undaria pinnatifida*, subcritical water extraction, bioactive compounds, antioxidant activity, antidiabetic potential, antihypertensive activity

## Abstract

The subcritical water extraction of *Undaria pinnatifida* (blade, sporophyll, and root) was evaluated to determine its chemical properties and biological activities. The extraction was conducted at 180 °C and 3 MPa. Root extracts exhibited the highest phenolic content (43.32 ± 0.19 mg phloroglucinol/g) and flavonoid content (31.54 ± 1.63 mg quercetin/g). Sporophyll extracts had the highest total sugar, reducing sugar, and protein content, with 97.35 ± 4.23 mg glucose/g, 56.44 ± 3.10 mg glucose/g, and 84.93 ± 2.82 mg bovine serum albumin (BSA)/g, respectively. The sporophyll contained the highest fucose (41.99%) and mannose (10.37%), whereas the blade had the highest galactose (48.57%) and glucose (17.27%) content. Sporophyll had the highest sulfate content (7.76%). Key compounds included sorbitol, glycerol, L-fucose, and palmitic acid. Root extracts contained the highest antioxidant activity, with IC_50_ values of 1.51 mg/mL (DPPH), 3.31 mg/mL (ABTS^+^), and 2.23 mg/mL (FRAP). The root extract exhibited significant α-glucosidase inhibitory activity with an IC_50_ of 5.07 mg/mL, indicating strong antidiabetic potential. The blade extract showed notable antihypertensive activity with an IC_50_ of 0.62 mg/mL. Hence, subcritical water extraction to obtain bioactive compounds from *U. pinnatifida*, supporting their use in functional foods, cosmetics, and pharmaceuticals is highlighted. This study uniquely demonstrates the variation in bioactive compound composition and bioactivities across different parts of *U. pinnatifida*, providing deeper insights. Significant correlations between chemical properties and biological activities emphasize the use of *U. pinnatifida* extracts for chronic conditions.

## 1. Introduction

Marine plants, such as seaweeds, are a valuable resource that provides humankind with food, nutrition, and pharmaceuticals. Of the diverse types of seaweeds, *Undaria pinnatifida* belongs to a group of brown seaweed native to China, Japan, and Korea, where it is grown commercially for food production, and is also available in the coastal areas of Russia, the United States, and some European countries [1]. *U. pinnatifida* is a significant resource with a global production exceeding 2.3 × 10^6^ tons, and >99% of this biomass originates from farming in 2018 [2]. It is primarily used for human consumption. It is also an additive to animal feed, biofuel production, fertilizer, and a source of bioactive compounds in nutraceuticals and cosmetics. *U. pinnatifida* contains diverse groups of bioactive compounds, such as polysaccharides, phenolics, flavonoids, phytosterols, peptides, omega-3 fatty acids, pigments, and vitamins [3]. Algae are rich in various bioactive compounds that contribute to their numerous health benefits. The major polysaccharides in brown algae include fucoidan, alginate, and laminarin. Fucoidan is renowned for its anticancer, anti-inflammatory, and antiviral properties. At the same time, alginate is widely utilized as a thickening agent in various industries and has significant biomedical applications, such as wound healing. Laminarin, a storage polysaccharide, is known for its antioxidant and immune-boosting effects. Brown algae also contain a diverse range of phenolic compounds, such as phlorotannins and flavonoids. Phlorotannins are unique to brown algae and possess strong antioxidant and anti-inflammatory properties. Flavonoids are known for their antioxidant and anti-inflammatory effects. Furthermore, brown algae are a valuable source of omega-3 fatty acids, such as EPA and DHA, which benefit cardiovascular health. Additionally, they contain fucoxanthin, a potent antioxidant that has been shown to aid in weight loss and possess anti-obesity properties. [4]. The compounds found in *U. pinnatifida* contain many remarkable pharmacological properties and health benefits, including immunomodulatory, free radical scavenging, anti-inflammatory, antitumor, antidiabetic, antimicrobial, antihypertensive, antithrombotic, and anti-obesity properties with fewer side effects and low toxicity [5,6,7]. It is generally considered a “longevity sea vegetable” used in many Western countries throughout history and has been used for the treatment of goiter, stomach ailments, urinary disease, dropsy, scrofula, and hemorrhoids in Southeast Asian countries [8,9].

Extraction techniques significantly influence not only yield, purity, and efficacy, but also production cost and biofunctional activity. Selecting a suitable extraction method with optimal operating parameters is important for obtaining bioactive compounds from seaweeds, enhancing their bioavailability and potential, and maximizing their therapeutic effects [10]. Traditional extraction techniques, including physical (mechanical grinding and centrifugation), chemical (organic solvents), and enzyme-based methods, are commonly used to prepare bioactive compounds from seaweeds; however, they have several drawbacks. Physical techniques are time-consuming and inefficient with respect to yield. Chemical extraction using organic solvents (e.g., methanol, ethanol, and n-hexane) is often used for seaweed extraction; however, the se solvents can negatively affect the content of the active ingredients, reduce extraction output, increase production expense, and are potential safety hazards because of residual solvent [11]. Therefore, it is necessary to develop efficient and green extraction techniques to address these limitations. Subcritical water extraction is an emerging, eco-friendly, and clean extraction process that occurs at temperature ranging from 100 to 374 °C and pressure (1–22 MPa) of water [12,13]. At room temperature, the dielectric constant (ε) of water is 80, which is a notable characteristic of subcritical water; however, this value decreases to 25 under subcritical conditions of 250 °C and 2.5 MPa. This characteristic dielectric constant (ε) of subcritical water is comparable with that of organic solvents, such as acetone (ε = 20.7), methanol (ε = 32.6), and ethanol (ε = 24.3). The polarity of subcritical water can be manipulated by altering temperature and pressure, which ultimately enables the selective extraction of bioactive compounds from a sample matrix [14].

Oxidation reactions in living organisms are unavoidable but necessary for energy production. This results in the formation of singlet oxygen (1O_2_), hydrogen peroxide (H_2_O_2_), and hydroxyl radicals (OH•), commonly known as reactive oxygen species (ROS). ROS produced from energy metabolism, stress, external chemicals, or food can oxidize cell molecules resulting in destructive and irreversible damage. The primary targets of this degradation process are DNA, proteins, cell membranes, and essential cellular components, which can lead to severe physiological disorders, such as heart problems, diabetes, muscular dystrophy, arthritis, neurological dysfunctions, cancer, and aging [15]. Therefore, it is essential to consume antioxidant compounds to neutralize ROS and protect the body. While synthetic antioxidant compounds are available, they can have lethal side effects. Therefore, health experts recommend natural antioxidants, such as extracts from the bioactive-rich *U. pinnatifida*. Its wealth of compounds can meet the current demand for functional foods and other products.

Utilizing waste from aquatic biomass is eco-friendly as it involves waste valorization, compensates for the intensive exploitation of natural stocks, meets nutritional demand, and protects the environment from unregulated waste disposal. Of the three major parts of *U. pinnatifida*, the blade is edible, whereas the sporophyll and root parts are considered waste. The waste portion, especially the sporophyll, has garnered interest because of its richness in bioactive compounds, including fucoidan, which is a unique polysaccharide [16]. This study focused on a comparative analysis of the various bioactive compounds and their activities in the three major parts of *U. pinnatifida* extracted by subcritical water hydrolysis. Other studies of *U. pinnatifida* are available that describe the bioactive compounds and their functions using conventional extraction techniques [5,6,7]. In addition, various seaweeds have been subjected to subcritical water extraction and have shown superior results compared with conventional extraction [10,17,18,19]. However, to our knowledge, no reports have described the subcritical extraction of bioactive compounds from *U. pinnatifida* and its various parts, including the blade, sporophyll, and root, using subcritical water extraction.

Therefore, we prepared bioactive extracts from the three major body parts of *U. pinnatifida* and evaluated their potential commercial value for use in functional foods, cosmetics, and pharmaceutical products. In addition to the physicochemical parameters, the content of functional compounds, such as bioactive polysaccharides, total proteins, phenols, and flavonoids, were determined, while simultaneously evaluating a wide range of biological properties, including free radical scavenging, antihypertensive, antibacterial, and antidiabetic activities.

## 2. Results and Discussion

### 2.1. Proximate Composition

The results of a proximate composition analysis for the various parts of *U. pinnatifida* are listed in Table 1. Excluding the roots, the blade and sporophyll sections had the highest carbohydrate content (39.45% ± 0.45% for the blade and 51.33% ± 0.39% for the sporophyll). The high carbohydrate content in *U. pinnatifida* suggests its potential as a rich source of polysaccharides. Specifically, the sporophyll is considered to have a higher carbohydrate content compared with the other parts because of its rich content of industrially important alginic acid and fucoidan [1]. The polysaccharides contained in *U. pinnatifida* possess various bioactive properties and have significant potential for applications in diverse industries. These polysaccharides are known for their antitumor, antibacterial, immunostimulatory, and anti-inflammatory effects [1,20]. Because of these characteristics, *U. pinnatifida* polysaccharides are considered valuable materials for drug delivery systems, tissue engineering, and skin regeneration. Studies indicate that the cell protective and regenerative effects of these polysaccharides may lead to innovative products in the medical and biotechnology sectors [21]. Unlike other parts, the roots of *U. pinnatifida* exhibited the highest ash content at 41.73% ± 0.73%, likely because the roots accumulate minerals from the environment while firmly attached to ropes or rocks in aquaculture settings. Although seaweeds generally absorb nutrients across their entire surface, the roots are physically attached to rocks or other fixed surfaces, which enables them to accumulate minerals. This characteristic explains the higher ash content detected in the root section.

### 2.2. Extraction Efficiency

The extraction efficiency varied by part and ranged from 65.93% ± 0.39% to 80.40% ± 0.65%, with the blade section showing the highest extraction efficiency and the root section having the lowest (Table 2). Statistical analysis using ANOVA indicated that the differences in extraction efficiencies among the different parts were statistically significant (*p* < 0.05). In previous research, the extraction efficiency of the blade was reported to be 81.88 ± 1.35% [10], which is very similar to the efficiency found in our study. However, the extraction efficiency of the sporophyll in previous research was 80.30 ± 0.02% [22], which is higher than the 73.87 ± 0.32% observed in our study. This discrepancy may be due to the differences in composition related to the harvest time of the sporophyll, which contains reproductive cells. Further studies are needed to investigate the composition changes in sporophyll at different harvest times. The higher extraction efficiency in the blade section could be attributed to its higher surface area and potentially higher water-soluble polysaccharides, which facilitate the extraction process. In contrast, the lower efficiency observed in the root section may be due to its denser structure and different composition, which includes higher amounts of fibrous materials that are less amenable to extraction. Differences in extraction yields under similar conditions may be attributed to the variations in the composition of the polysaccharides, proteins, ash, and other compounds present in the different parts. Further studies are needed to fully understand these results.

### 2.3. Color and Maillard Reaction Products (MRPs)

The colorimetric analysis results for *U. pinnatifida* subcritical water extracts (USE-s) are listed in Table 2. The lightness value (L*) varied from 27.68 ± 0.87 to 32.30 ± 0.73. The a* values (red to green) ranged from 11.56 ± 0.44 to 14.00 ± 0.45, and the b* values (yellow to blue) from 6.96 ± 0.56 to 13.65 ± 1.54. The USE-B showed higher L* and b* values, whereas the USE-R exhibited lower values overall. The chroma (C*) and hue angle (H°) values further elucidate the color characteristics of the extracts. The C* values ranged from 13.28 ± 0.56 to 19.07 ± 1.96, indicating varying degrees of color saturation among the different parts of the seaweed. Higher C* values, as observed in USE-B, suggest a more vivid and intense color, corresponding with the higher b* values noted previously. The hue angle (H°) values varied from 30.67 ± 1.40 to 45.65 ± 1.16, showing differences in the perceived color tone. A hue angle of 45.65° in USE-B indicates a more orange hue, while lower hue angles in USE-S (36.27°) and USE-R (30.67°) shift towards a more yellow–orange and red–orange hue, respectively. Overall, the colorimetric data, including L*, a*, b*, C*, and H° values, demonstrate that the specific part of the seaweed used for the extraction significantly influences the color properties of the resulting extracts. The higher L*, b*, and C* values in USE-B can be attributed to the higher concentrations of pigments such as β-carotene and fucoxanthin in the blade and sporophyll, contributing to the vivid and intense coloration. The variations in hue angle further highlight the differences in pigment composition and distribution across the different parts of *U. pinnatifida* [23].

MRPs form when carbonyl groups in reducing sugars and free amino acids react under high-temperature conditions, such as subcritical water treatment [24]. MRPs are compounds that significantly influence food quality attributes, such as taste, aroma, color, and texture. They are formed through non-enzymatic reactions that occur between proteins or amino acids and sugars, particularly in heat-processed foods. The formation of MRPs is observed at two major wavelengths: 294 nm and 420 nm. Absorbance at 294 nm indicates the presence of early and intermediate MRP compounds, which are used to monitor reaction progress. In contrast, absorbance at 420 nm reflects the browning intensity caused by high-molecular-weight compounds, such as melanoidins, which is useful for evaluating the color of the final products [25]. The ratio of these two absorbance values (A294/A420) indicates the efficiency of UV absorption, the conversion of UV-absorbing substances into other polymers, and the browning intensity [26]. The results of the MRP analysis are listed in Table 2. The mean UV absorbance (Abs294), browning intensity (Abs420), and the absorbance ratio (Abs294/420) ranged from 2.580 ± 0.050 to 2.886 ± 0.062, 0.158 ± 0.004 to 0.225 ± 0.003, and 12.844 ± 0.188 to 18.185 ± 0.279, respectively. Statistical analysis using ANOVA indicated that the differences in the Abs294, Abs420, and Abs294/420 values among the different parts were statistically significant (*p* < 0.05). These significant differences highlight the varying degrees of Maillard reaction product formation across the different parts of the seaweed. For MRPs obtained from the seaweed extracts, notable improvements were observed in functional properties, such as solubility, emulsification, foaming capacity, surface hydrophobicity, and antioxidant characteristics, including radical scavenging as determined by DPPH and CUPRAC analyses [27].

### 2.4. Total Phenolic Contents (TPC) and Total Flavonoid Contents (TFC)

The total phenolic and total flavonoid contents of the USE-s are listed in Table 3. Phlorotannins are tannins uniquely found in brown seaweed and are known for their potent antioxidant, anti-inflammatory, and antibacterial effects [28]. Although research on flavonoids in seaweeds is not as extensive as that in terrestrial plants, various studies have indicated the presence of functional flavonoids in seaweeds, such as quercetin, catechol, and rutin [29].

The determination of phenolic content revealed that USE-R had the highest concentration at 43.32 ± 0.19 mg PGE/g of dry sample, which was statistically significant (*p* < 0.05). In contrast, USE-B and USE-S exhibited significantly lower concentrations at 33.13 ± 0.14 and 30.11 ± 0.35 mg PGE/g of dry sample, respectively. The flavonoid content was also the highest in USE-R (31.54 ± 1.63 mg QE/g), significantly higher (*p* < 0.05) than that of USE-B (19.91 ± 0.54 mg QE/g) and USE-S (9.22 ± 0.54 mg QE/g). Previous studies have mostly focused on the overall phenolic content in seaweeds without differentiating between the various parts of the same species. However, our findings indicate that the roots of *U. pinnatifida*, which are directly exposed to more severe environmental stressors compared to other parts, exhibit significantly higher concentrations of phenolic compounds. The results suggest that the roots of *U. pinnatifida* may be valuable for use in functional foods and pharmaceuticals. The roots, which are the part where the seaweed attaches and grows, are exposed to various environmental stresses, such as hypoxia, salinity changes, and physical damage, which may lead to the increased production of compounds, including phlorotannins and flavonoids, as defensive mechanisms against these stress factors [30]. In previous research, the TPC of subcritical water extracts from New Zealand *U. pinnatifida* was found to be up to 29.9 ± 1.8 mg gallic acid equivalent/g dried mass [31]. Similarly, the subcritical water extraction *of U. pinnatifida* sporophyll from Korea showed TPC values up to 38.04 ± 0.18 PG E/g and TFC of 10.34 ± 0.24 mg QE/g [22]. Based on this study, it is evident that different parts of *U. pinnatifida* hold potential as functional materials. To enhance the value of each part as a functional ingredient, further studies are needed to isolate and structurally analyze phenolic and flavonoids from each part. In addition, functional evaluation and studies on the molecular and in vivo action mechanisms are needed.

### 2.5. Total Sugar Content (TSC), Reducing Sugar Content (RSC), and Total Protein Content (TPrC)

The TSC, RSC, and TPrC of the USE-s are listed in Table 3. The TSC was highest in USE-S (97.35 ± 4.23 mg glucose/g), which was significantly higher compared with that in USE-B (36.43 ± 0.75 mg glucose/g) and USE-R (57.04 ± 1.39 mg glucose/g). Similarly, the RSC was also highest in USE-S (56.44 ± 3.10 mg glucose/g) compared with USE-B (21.33 ± 0.51 mg glucose/g) and USE-R (39.44 ± 3.61 mg glucose/g). This indicates that the sporophyll of *U. pinnatifida* is rich in sugars, which suggests its potential as an energy source. The high sugar content in the sporophyll is particularly related to the presence of functional polysaccharides, such as fucoidan and alginate. Fucoidan is a sulfated polysaccharide primarily found in seaweeds and is known for its various bioactivities, including anticancer, anti-inflammatory, and antiviral properties. Alginate, a major polysaccharide found in brown seaweed, is used in the food and pharmaceutical industries because of its excellent viscosity and gel-forming properties. The high content of fucoidan and alginate in the sporophyll enhances its potential as a functional food ingredient and biomaterial [32]. In previous research, the TSC of the subcritical water-treated sporophyll of *U. pinnatifida* at 180 °C was found to be 81.84 ± 2.27 mg glucose/g, and RSC was 53.15 ± 0.50 mg glucose/g, which is consistent with the high sugar content observed in our study [22]. In addition, when subcritical water was applied from *Ecklonia stolonifera*, a type of brown seaweed, TSC was confirmed to be up to 102.04 ± 1.07 mg glucose/g and RSC was confirmed to be 61.83 ± 2.44 mg glucose/g [33].

TPrC was highest in USE-B (83.47 ± 1.76 mg BSA/g) compared with USE-S (84.93 ± 2.82 mg BSA/g) and USE-R (65.91 ± 3.53 mg BSA/g). Again, the statistical analysis using ANOVA indicated significant differences among the parts (*p* < 0.05). This suggests that the blades of *U. pinnatifida* are a significant source of protein, which highlights the nutritional value of the blade extracts. Proteins perform various functions within the body, and water-soluble peptides hydrolyzed and isolated from natural marine proteins exhibit antitumor, antidiabetic, and antihypertensive effects [34]. Recently, various functional foods containing seaweed-derived peptides have been commercialized [35,36]. Additionally, TPrC extracted from red algae such as Porphyra (laver) using subcritical water at 180 °C was found to be 20.21 ± 0.27 g BSA/100 g, demonstrating the potential of subcritical water extraction for obtaining high protein yields from marine sources [14].

The significant differences in TSC, RSC, and TPrC among the different parts of *U. pinnatifida* highlight the importance of considering the specific part of the seaweed used for extraction processes. Further research is needed to fully understand the factors affecting these differences and optimize extraction methods for maximizing the functional properties of seaweed-derived compounds.

### 2.6. Monosaccharide Composition, Sulfate Content, and Molecular Weight Analysis

Fucoidan is a sulfated polysaccharide primarily composed of α-1,3 and α-1,4 linked L-fucose residues. It belongs to both homopolysaccharides and heteropolysaccharides. In addition, it was found to contain D-galactose, D-mannose, D-xylose, L-rhamnose, D-glucuronic acid residues, and acetyl groups as components. These polysaccharides exhibit various bioactivities depending on their structure [37].

In the present study, we analyzed the monosaccharide composition and sulfate content of the USE-s to determine the bioactivity of functional polysaccharides such as fucoidan (Table 4). The main monosaccharides identified were fucose, galactose, glucose, and mannose, with some variation by part. The major monosaccharides in USE-B were galactose (48.57% ± 0.27%) and glucose (17.27% ± 0.73%). In USE-S, fucose (41.99% ± 0.09%) and mannose (10.37% ± 3.23%) were predominant, whereas in USE-R, the composition was primarily fucose (25.65% ± 2.25%) and mannose (17.40% ± 0.60%). These results are consistent with the previous analyses of the monosaccharide composition in the extracts of *U. pinnatifida* [10,38,39]. The statistical analysis using ANOVA indicated that the differences in monosaccharide compositions among the different parts were statistically significant (*p* < 0.05). These significant differences suggest that the specific part of the seaweed influences the monosaccharide profile, impacting the bioactivity of the extracts.

The sulfate content significantly affects the bioactivity of functional polysaccharides, such as fucoidan, carrageenans, and ulvan [40]. The results indicated that the sulfate content of USE-S (7.76% ± 0.17%) was the highest, significantly surpassing that of USE-B (2.50% ± 0.10%) and USE-R (2.41% ± 0.20%). High sulfate content enhances the various bioactivities of the functional polysaccharides, including anticoagulant, anticancer, and antiviral effects [41]. This study indicates that the sporophyll has considerable potential as an excellent functional material. In previous research, applying subcritical water extraction to Ecklonia maxima at similar conditions of 180 °C for 23.75 min resulted in a sulfate content of 12.06 ± 0.02 mg SO_4_^2−^/g [19]. For the fucoidan extracted from Saccharina japonica using subcritical water, the sulfate content was found to be as high as 28.64%, which is significantly higher than the sulfate content obtained in this study [42]. This suggests that further purification steps could enhance the sulfate content in our fucoidan extracts.

Molecular weight is an important indicator of biological activity [43]. In addition, using subcritical water extraction facilitates the modulation of the molecular weight of natural substances, including polysaccharides [44]. The molecular weight analysis of USE-s revealed a diverse distribution across the different parts. For example, USE-B exhibited a range of Mn from 192 to 2263 Da and Mw from 208 to 2924 Da, indicating the presence of natural substances of various sizes. USE-S and USE-R also showed a similar distribution. Generally, natural substances with higher molecular weights have stronger immunomodulatory and anticancer effects, whereas those with lower molecular weights have higher absorption rates, enabling rapid action within the body [45]. Thus, by adjusting the molecular weight through subcritical water extraction, it is possible to obtain functional materials with the desired bioactivities.

### 2.7. GC–MS Analysis

GC–MS analysis tentatively identified various compounds in the USE-s (Table 5). The identified compounds include sugar alcohols, monosaccharides, fatty acids, fatty amides, amino acids, glycosides, and monoglycerides. Their bioactivity and potential applications are described below.

#### 2.7.1. Sugar Alcohols and Monosaccharides

The major sugar alcohol identified in the USE-s was sorbitol, which was present in high proportions in USE-B (53.57%), USE-S (49.55%), and USE-R (33.82%). Sorbitol has significant potential as a sweetener, humectant, and laxative, making it useful for many industries. Notably, sorbitol may be used as a sugar substitute because of its low glycemic index, making it suitable for diabetic patients. In addition, its low risk of causing dental cavities makes it a popular ingredient in dental products [46]. Glycerol, which is present at 1.55% in USE-B and 1.08% in USE-S, is an important ingredient in the cosmetics industry as a humectant and skin protectant. Its high moisture-retaining capacity maintains skin hydration and provides a protective barrier against external factors [47]. The primary monosaccharide identified was L-fucose (2.93% in USE-S), known for its anti-inflammatory and anticancer effects, particularly in modulating the immune response and reducing inflammation [48]. D-galactose is present at 0.47% in USE-S and 1.11% in USE-R and acts as a prebiotic to promote the growth of beneficial gut bacteria and contributes to improved gut health [49].

#### 2.7.2. Fatty Acids and Fatty Amides

Palmitic acid (6.79% in USE-B, 4.25% in USE-S, and 11.49% in USE-R) plays an important role in skin health by strengthening the skin barrier and providing anti-inflammatory and antibacterial effects. This fatty acid also contributes to energy storage and cell membrane composition [50]. Stearic acid (4.74% in USE-B, 2.80% in USE-S, and 7.73% in USE-R) is used as a moisturizer and stabilizer, forming a protective barrier on the skin to prevent moisture loss and maintain soft skin. In addition, stearic acid serves as an emulsifier in the cosmetics and pharmaceutical industries [51]. 9-Octadecenamide (1.46% in USE-S) is known for its potential role as an endogenous sleep-inducing mechanism in mammals. One of the most potent physiological effects of oleamide is vasodilation, which is mediated through endothelial-derived nitric oxide, endothelium-dependent hyperpolarization, and TRPV1 receptor activation. Furthermore, oleamide exhibits anticonvulsant activity and significantly reduces the severity of seizures induced by pentylenetetrazole [52].

#### 2.7.3. Amino Acids

D, L-Pyroglutamic acid (2.15% in USE-B and 1.79% in USE-S) has moisturizing and antioxidant effects. It acts as a natural moisturizing factor and plays an important role in maintaining and protecting skin hydration [53]. L-aspartic acid (0.49% in USE-B, 0.50% in USE-S) is vital for energy metabolism, particularly in ATP synthesis. In addition, aspartic acid is involved in the synthesis of neurotransmitters, which support brain function [54].

#### 2.7.4. Glycosides and Monoglycerides

Glyceryl glycoside (2.69% in USE-B, 0.27% in USE-S, and 0.27% in USE-R) is a product of the reaction between glucose and glycerin. Recent studies suggest that glyceryl glucoside plays a key role in inducing aquaporin-3 protein, which is essential for water molecule transport across the skin cell membrane. As a result, glyceryl glucoside is emerging as a next-generation moisturizing ingredient [55]. 1-Monopalmitin (2.40% in USE-S) has a potential role as an inhibitor of P-glycoprotein (P-gp). P-gp is an ATP-binding cassette transporter that plays an important role in drug efflux and multidrug resistance. The inhibition of P-gp by 1-Monopalmitin enhances drug bioavailability and is significant in reversing drug resistance in cancer [56].

### 2.8. Biological Activity

#### 2.8.1. Antioxidant Activity

Seaweeds are rich in compounds with antioxidant activity and may be used as a novel source of antioxidants for the food, feed, and cosmetics industries [57]. To measure the antioxidant activity of USE-s, we determined the 2,2′-azino-bis(3-ethylbenzothiazoline-6-sulfonic acid) (ABTS^+^) and 2,2-Diphenyl-1-picrylhydrazyl (DPPH) radical scavenging abilities as an IC_50_ and Ferric Reducing Antioxidant Power (FRAP) reducing power as an EC_50_. USE-R exhibited the highest activity with values of 1.51 ± 0.12, 3.31 ± 0.24, and 2.23 ± 0.13 mg/mL, respectively (Table 6). This high activity was likely affected by the high TPC (43.32 ± 0.19 mg PGE/g) and TFC (31.54 ± 1.63 mg QE/g) content in USE-R. Phenolics and flavonoids, which are abundant in brown seaweeds, act as powerful antioxidants and contain significantly higher polyphenol content and potential antioxidant capacity compared with green and red seaweeds [58]. Among the different parts, USE-R had a higher content of phenolic compounds; however, further studies using advanced analytical methods, such as UPLC-MS/MS, are necessary to characterize the individual phenolic compounds present in the extracts from each part.

To confirm the correlation between phenolic compounds and antioxidant activity, we analyzed the relationship between their chemical properties and biological activities (Table 7). The correlations between TPC and ABTS^+^, DPPH, and FRAP were *R*^2^ = −0.925, −0.941, and −0.992, respectively. The correlations between TFC and ABTS^+^, DPPH, and FRAP were *R*^2^ = −0.994, −0.998, and −0.998, respectively, indicating a strong correlation between phenolic compounds and antioxidant activity. Previous studies have also demonstrated a strong correlation between TPC, TFC, and antioxidant activity [10]. Previous research has also confirmed strong correlations between antioxidant activity and phenolic/flavonoid content in different seaweeds. For instance, in red algae *Gracilaria changii*, strong positive correlations were found between TPC and DPPH (*R*^2^ = 0.999) and FRAP (R^2^ = 0.994), as well as between TFC and DPPH (R^2^ = 0.994) and FRAP (*R*^2^ = 0.989) [59]. In brown algae *Sargassum thunbergii* extracts, strong positive correlations were observed between TPC and ABTS^+^ (*R*^2^ = 0.968), DPPH (*R*^2^ = 0.965), and FRAP (*R*^2^ = 0.984) [60].

#### 2.8.2. α-Glucosidase Inhibitory

Diabetes is a chronic disease characterized by abnormally high blood sugar levels resulting from problems with insulin secretion or action. Over time, it can cause serious complications, such as cardiovascular disease, kidney disease, and vision impairment. The primary goal of diabetes management is to maintain blood glucose levels within a normal range utilizing dietary therapy, exercise, and medication [61]. α-Glucosidase is an enzyme in the small intestine that breaks down carbohydrates into monosaccharides and directly affects blood sugar elevation. Inhibiting the activity of this enzyme can delay the digestion and absorption of carbohydrates, thereby reducing postprandial blood glucose spikes. Natural α-glucosidase inhibitors are gaining attention as alternative therapies because they have fewer side effects and greater safety for long-term use compared with synthetic drugs [62].

The antidiabetic activity results expressed as IC_50_ values for α-glucosidase inhibitory activity are shown in Table 6. USE-B exhibited an IC_50_ of 17.52 ± 0.71 mg/mL, USE-S had an IC_50_ of 14.69 ± 0.59 mg/mL, whereas USE-R showed the highest inhibitory activity with an IC_50_ of 5.07 ± 0.45 mg/mL. A high correlation between α-glucosidase inhibitory activity and chemical properties with TPC was observed (*R*^2^ = −0.905). In contrast, TPrC showed a strong negative correlation (*R*^2^ = 0.959) (Table 7). Various studies have identified fatty acids, phenols, and terpenes as major α-glucosidase inhibitors (AGIs). A molecular docking analysis of poricoic acid A and quercetin-3-O-glucuronide extracted from brown seaweed revealed that these compounds form more than four hydrogen bonds and bind to 2–4 active sites in α-glucosidase, confirming their potential as AGIs [63]. Further studies are needed to profile the metabolites in USE-s and identify superior AGIs.

#### 2.8.3. Antihypertensive Activity

A key treatment method for managing hypertension is the inhibition of Angiotensin-Converting Enzyme (ACE). ACE inhibitors act by blocking the conversion of angiotensin I to angiotensin II, which relaxes blood vessels and lowers blood pressure. Pharmaceutical companies have produced various ACE inhibitors to reduce angiotensin II levels and, consequently, manage hypertension; however, these drugs often come with a range of undesirable side effects, highlighting the need for natural, food-derived ACE inhibitors that control hypertension with minimal side effects [64].

The antihypertensive activity was evaluated by measuring ACE inhibitory activity, which is expressed as an IC_50_ value. USE-B had an IC_50_ of 0.62 ± 0.01 mg/mL, USE-S had an IC_50_ of 0.76 ± 0.01 mg/mL, and USE-R had an IC_50_ of 0.90 ± 0.02 mg/mL. The standard compound captopril exhibited an IC_50_ of 4.35 × 10^−5^ ± 0.01 mg/mL. These results indicate that USE-B has the highest ACE inhibitory activity, followed by USE-S and USE-R. The antihypertensive activity showed a strong correlation with TPrC, with an R^2^ value of −0.829 (Table 7). This suggests that the high antihypertensive activity of USE-s is related to their high protein content. Peptides are widely studied natural compounds for inhibiting ACE1 activity [65]. In previous studies, water-soluble proteins (WSPs) derived from the red seeed *Pyropia pseudolinearis*, containing phycoerythrin, phycocyanin, allophycocyanin, and ribulose 1,5-bisphosphate carboxylase/oxygenase-derived peptides, have shown excellent antihypertensive activity [66].

The examples of commercially available antihypertensive peptides derived from macroalgae and approved by Japan’s Ministry of Health, Labor, and Welfare as FOSHU (“Food for Specified Health USE-s”) include Ameal-S 120^®^ (Calpis Co., Ltd., Tokyo, Japan) and Evolus^®^ (Valio Ltd., Helsinki, Finland) [67]. Further studies are needed to elucidate the structure and antihypertensive mechanisms of peptides derived from each extract and to develop efficient methods for peptide conversion. This will provide a basis for the development of natural antihypertensive agents using seaweed.

### 2.9. Principal Component Analysis

Principal component analysis (PCA) involves a mathematical procedure that identifies patterns in a data set and then expresses the data in such a way as to highlight their similarities and differences. This study analyzed the relationship between chemical properties and biological activities using PCA (Figure 1). The plot shows that components 1 and 2 explain most data variance. The observations from the PCA biplot largely align with those from Pearson correlation studies. PCA achieved a total variance explanation of 98.38%, with the first and second principal components accounting for 71.14% and 27.24% of the variance, respectively. Notably, TPC and TFC have negative values in components 1 and 2, suggesting they share similar characteristics. In contrast, RSC and TSC exhibit very high positive values in component 2, indicating that these chemical properties tend to vary. Additionally, variables such as ABTS^+^, DPPH, FRAP, and α-glucosidase inhibitory have positive values in component 2. The fact that these variables show an opposite variance pattern compared to TPC and TFC is likely due to the IC_50_ method used, where lower values indicate higher biological activity. This also explains why TPrC and antihypertensive activities are positioned oppositely. Previous studies using subcritical water to assess TPC and antioxidant activity have yielded similar results [68,69]. While this study has elucidated the relationship between components and activities through overall analysis, further research is needed to increase the reliability of the analysis by including more samples and components. Additionally, further studies using different analytical techniques are required to clarify the relationships between variables.

## 3. Materials and Methods

### 3.1. Materials and Chemicals

*U. pinnatifida*, cultivated in the Gijang area of Busan, was purchased from Dndn-Bada (Busan, Republic of Korea, 35.187753° N, 129.211470° E). The roots of *U. pinnatifida* were collected as by-products during the harvesting stage, whereas the blades and sporophylls were separated during the processing stage. Each part was thoroughly cleaned with fresh water to remove non-target materials, salts, and minerals. The samples were dried at 40 °C for 72 h, processed using a PN SMKA-4000 mixer (PN Poong Nyun Co., Ltd., Ansan-si, Republic of Korea), and sieved through a 710 μm mesh to ensure a uniform particle size. The resulting seaweed powders were stored at −40 °C in sealed containers for future use. To obtain the target pressure in subcritical water, 99.99% pure nitrogen gas was obtained from KOSEM (Busan, Republic of Korea). The standards were purchased from Sigma-Aldrich (St. Louis, MI, USA), whereas the chemicals and reagents were obtained from Samchun Pure Chemical Co., Ltd. (Pyeongtaek, Republic of Korea).

### 3.2. Proximate Composition

The proximate composition of the blade, sporophyll, and root of *U. pinnatifida* was analyzed following the Association of Official Agricultural Chemists methods as previously described [60].

### 3.3. Subcritical Water Extraction

A batch-type stainless steel reactor (Phosentech, Daejeon, Republic of Korea) was used for subcritical water extraction following a previously reported method with minor modifications [10]. In the 500 cm^3^ reactor, 15 g of the seaweed sample was mixed with double-distilled water at a ratio of 1:20 (*w*/*v*). The pressure was maintained at 3 MPa using nitrogen gas, and the extraction was conducted at 180 °C for 30 min. During extraction, the solid–solvent mixtures were stirred at 200 rpm using a double, four-blade impeller. The average heating times for the blade, sporophyll, and root were 24 min, 22 min, and 22 min, respectively. The resulting *U. pinnatifida* subcritical water extract was designated as USE. The extracts from the blade, sporophyll, and root were designated USE-B, USE-S, and USE-R, respectively.

After the reaction, USE-s were collected, filtered using a filter paper (CHMLAB GROUP, F1091-110), and lyophilized and stored at −70 °C. For the subsequent experiments, the freeze-dried extract powder was appropriately diluted with distilled water to match the standard curves for each assay. To ensure accuracy, the experiments were conducted at multiple concentrations to confirm the consistency of the results.

The solid–solvent mixture was immediately cooled using the chilled water line attached to the reactor. The USE-s were filtered using the CHMLAB GROUP filter paper (F1091-110) and stored at 4 °C. The extraction efficiency was calculated based on the weight of the solid residue after drying at 55 °C for 48 h to ensure complete moisture removal using the following formula:Extraction efficiency (%) = (W − W1/W) × 100
where W stands for the original sample taken for hydrolysis, and W1 is the weight of the dried solid extract after hydrolysis.

### 3.4. Chemical Properties of USE

#### 3.4.1. Color, pH, and MRPs

To determine USE color, different parameters (L*, a*, and b* values) were measured using a colorimeter (Lovibond RT series, The Tintometer Ltd., Amesbury, UK). The values of L* varied from 0 to 100, indicating the lightness of the extracts, where a* denotes red to green and b* yellow to blue. The standard plate values were L* = 94.92, a* = −1.04, and b* = 0.19. From these, the chroma (C* = 1.057) and hue angle (h* = 169.65°) were calculated. Each measurement was conducted three times, and the average was calculated.

The pH of the USE-s was measured at room temperature using a pH meter (Thermo Scientific, Waltham, MA, USA, ORION STAR A211). Before the measurement, the pH meter was calibrated using buffer solutions of pH 2, 4, 7, and 10.

The UV absorbance and browning of the MRP samples were measured according to previously described methods [10]. Each hydrolysate was diluted 20× with distilled water and the absorbance was measured at 294 and 420 nm using a Synergy HT microplate reader (BioTek Instruments, Winooski, VT, USA). The absorbance ratio (A294/A420) was calculated to monitor the conversion of UV-absorbing compounds into brown polymers.

#### 3.4.2. Total Phenolic and Total Flavonoid Contents

The TPC and TFC of the USE-s were analyzed using modified standard methods. For TPC, the procedure described by Park et al. (2022) [60] was followed with minor adjustments. USE-s were mixed with Folin–Ciocalteu phenol reagent and sodium carbonate. The absorbance was measured at 765 nm using phloroglucinol as a standard. The results were expressed as mg phloroglucinol equivalent per gram of dry sample (mg PGE/g of dry sample). For TFC, the method was adapted from Kim et al. (2024) [22] and included a reaction with sodium nitrite, aluminum chloride, and sodium hydroxide, with the absorbance read at 510 nm and quercetin as the standard. The results were expressed as mg quercetin equivalent per g of sample (mg QE/g of dry sample).

#### 3.4.3. Total Sugar and Reducing Sugar Content

The TSC and RSC of the USE-s were quantitated using previously described methods. TSC was measured using a protocol modified from Chamika et al. (2021) [70] involving a reaction with sulfuric acid and phenol, followed by absorbance measurement at 490 nm using a glucose standard curve for quantitation (mg glucose/g of dry sample.). For RSC, the 3,5-dinitrosalicylic (DNS) colorimetric assay was used as described by Ali et al. (2023) [44], with modifications to the DNS solution preparation and the assay conditions. Absorbance for RSC was measured at 540 nm and the results were compared against a glucose standard to express the values in mg glucose/g of dry sample.

#### 3.4.4. Total Protein Content

The TPrC of the USE-s was quantified using a modified Lowry method as detailed in Jeong et al. [71]. The solutions for the assay were prepared by dissolving NaOH, Na_2_CO_3_, potassium sodium-tartrate, and cupric sulfate in specified proportions. The assay mixture combined these solutions with the extract and Lowry’s reagent, followed by sequential dark incubations before and after the addition of the Folin–Ciocalteu reagent. The absorbance was measured at 750 nm using a microplate reader with bovine serum albumin (BSA) as the calibration standard. The results are expressed as mg BSA/g of dry sample.

#### 3.4.5. Monosaccharide Analysis

The monosaccharide composition of the USE-s was determined using an ion chromatograph (ICS-5000; Dionex, Sunnyvale, CA, USA) equipped with pulsed amperometric detection and a CarboPac_SA10G column (4 mm × 50 mm, Dionex, USA). Lyophilized USE powder (50 mg) was dissolved in a mixture of 5 mL 12 M sulfuric acid and 25 mL water, then hydrolyzed at 120 °C for 2 h for conversion to monosaccharides. The resulting hydrolysate was filtered through a 0.2 μm syringe filter. The high-performance liquid chromatography (HPLC) conditions were as follows: solvent A was deionized water and solvent B was 100 mM NaOH. The gradient program was 0–24 min in 6% B, 24–25 min transitioning from 6% to 100% B, 25–30 min in 100% B, 30–31 min returning from 100% to 6% B, and 31–50 min in 6% B. The flow rate was set at 0.6 mL/min with an injection volume of 10 μL.

#### 3.4.6. Sulfate Content

The sulfate content was determined turbidimetrically utilizing the BaCl_2_-gelatin method following hydrolysis in 0.5 M HCl, with slight modifications to the previously described procedure [72]. The lyophilized USE-s (5 mg) were hydrolyzed in 1 mL of 1 M HCl at 105 °C for 5 h. Once cooled, the solutions were thoroughly mixed and filtered through Whatman No. 1 filter paper. A 0.2 mL aliquot of the filtered solution was transferred to a 10 mL tube containing 3.8 mL of 3% trichloroacetic acid and 1 mL of BaCl_2_-gelatin reagent. The mixture was then vigorously shaken and incubated at room temperature for 15 min. The absorbance was measured at 360 nm. A blank was prepared with distilled water following the same procedure. A calibration curve was generated using K_2_SO_4_ at concentrations ranging from 0.06 to 0.6 mg/mL (approximately 0.0533–0.533 mg SO_4_^2−^/mL). The sulfate content was calculated and expressed as a percentage based on the dry weight.

#### 3.4.7. Molecular Weight Analysis

The molecular weight of the USE-s was assessed via gel permeation chromatography (GPC) using the HPLC Ultimate3000 RI System from Thermo Dionex, USA. The lyophilized USE-s were dissolved in deionized water at a concentration of 10 mg/mL. The solution was filtered using a 0.45 μm PTFE-H syringe filter and subsequently degassed. The GPC setup included an injection volume of 10 μL and a 0.1 M NaN3 aqueous solution as the mobile phase at a 1 mL/min rate. Water ultrahydrogel columns with pore sizes of 120, 500, and 1000 were connected in a series. Molecular weights were determined based on a calibration curve derived from pullulan standards ranging from 3.42 × 10^2^ to 8.05 × 10⁵ g/mol. The Chromeleon 6.8 expansion pack software (Thermo, USA) was used for data analysis.

#### 3.4.8. GC–MS

##### Sample Preparation

For the GC–MS analysis, 1 mg/mL USE-s were dissolved in HPLC-grade methanol. The solution was vortexed for 2 min and the extract was filtered through a 0.2 μm filter. Subsequently, 1 mL of the extract (25 g/mL) was placed into a GC vial for analysis. Methanol was completely evaporated using N_2_ gas. The sample underwent derivatization with 30 μL pyridine and 60 μL BSTFA: TMCS (99:1) by incubating at 60 °C for 60 min. Finally, 1 μL of the derivatized sample was used for the GC–MS analysis.

##### Analysis

The GC–MS measurements were carried out on a Shimadzu equipment (GCMS-QP-2020 NX, Shimadzu Corporation, Kyoto, Japan) equipped with electron impact ionization and single quadruple mass. The data were acquired using the GCMS solution (v4.54) software. The parameters included the following: column: DB-5MS, dimensions: 30 m length × 0.25 mm ID × 0.25 μm film thickness, 250 °C injection temperature, 50 °C primary temperature for 2 min hold time, 10–280 °C ram temperature for 5 min hold time, and 5–300 °C ram temperature for a 6 min hold time (40 min total run time). Helium was used as the carrier gas, the flow rate was 1.0 mL/min, the split ratio was 50 or 100, a 1 μL sample was injected, and the scan mass range was m/z 40–600, with positive polarity. The resulting spectra were compared with those of known compounds using the Wiley Registry^®^ 12th Edition/NIST 2020 Mass Spectral Library (2020) [73].

### 3.5. Biological Activity

#### 3.5.1. Antioxidant Activity (DPPH, ABTS^+^, and FRAP Assay)

The radical scavenging activities of the USE-s were measured using DPPH, ABTS^+^, and FRAP assays as previously described [74], with minor modifications. The absorbance of the reaction mixture was recorded three times at 517 nm (for the DPPH assay), 734 nm (for the ABTS^+^ assay), and 593 nm (for the FRAP assay) using a Synergy HT microplate reader (BioTek Instruments, Winooski, VT, USA). Trolox was used as a reference material. The antioxidant activity was calculated using the following formula and presented as a percentage (%):Antioxidant activity (%) = (A_blank_ − (A_sample_ − A_background_))/A_blank_ × 100
where A_blank_, A_background_, and A_sample_ correspond to the absorbance of the blank control, background control, and sample, respectively.

#### 3.5.2. Antihypertensive Activity

The ACE inhibitory activity of USEs was determined following the ACE kit-WST manual (Dojindo Molecular Technologies, Inc., Tokyo, Japan) [75]. The enzyme working solution, indicator working solution, and sample solution preparation are detailed in the Appendix A. To determine ACE inhibitory activity, we followed these steps. First, we added 20 µL of the sample solution to the designated sample wells in a 96-well microplate. For the blank1 and blank2 wells, we added 20 µL of deionized water. Next, we added 20 µL of substrate buffer to each sample well and to the blank1 and blank2 wells. We also added an additional 20 µL of deionized water to the blank2 wells. After that, we added 20 µL of the enzyme working solution to each sample well and the blank1 wells. We incubated the microplate at 37 °C for 1 h. Following the incubation, we added 200 µL of the indicator working solution to each well and incubated the plate at room temperature for 10 min. We then measured the absorbance at 450 nm using a microplate reader. We calculated the ACE inhibitory activity using the following formula and presented it as a percentage (%):ACE inhibitory activity (%) = (A_blank1_ − (A_sample_ − A_background_))/(A_blank1_ − A_blank2_) × 100
where A_blank1_ is the absorbance of the positive control without ACE inhibition, A_blank2_ is the absorbance of the reagent blank, A_background_ the absorbance using distilled water instead of enzyme and indicator in the sample, and A_sample_ is the sample absorbance.

#### 3.5.3. α-Glucosidase Inhibitory Activity

The α-glucosidase inhibitory activity was measured following a previously described method with slight modifications [76]. Briefly, various concentrations of the sample solution (0–50 mg/mL) were mixed with potassium phosphate buffer (200 mM, pH 6.8, 50 µL) and α-glucosidase (0.2 U/mL, 50 µL) in a 96-well plate and pre-incubated at 37 °C for 10 min. After the pre-incubation, pNPG (3 mM, 100 μL) was added and the reaction proceeded for another 10 min. The reaction was stopped by adding Na_2_CO_3_ (0.2 M, 750 μL), and the absorbance was measured at 405 nm using acarbose as the reference material. All the reagents and samples were dissolved in potassium phosphate buffer (200 mM, pH 6.8). The samples and reaction systems without added enzymes were used as blank and background controls, respectively.

### 3.6. Statistical Analysis

The values are expressed as the mean ± standard deviation of triplicate determinations. Statistical analyses (one-way analysis of variance, Pearson’s correlation analysis, and principal component analysis) were performed using the SPSS software (version 27, SPSS Inc., Chicago, IL, USA). Tukey’s HSD multiple range test was used to determine significant differences between the mean values at *p* < 0.05.

## 4. Conclusions

In this study, the various parts of *U. pinnatifida*, namely the blade, sporophyll, and root, were utilized to obtain extracts by SWH for the complete valorization and utilization of the biomass, highlighting bioactive compounds and their potential as sources of functional materials. The USE-B exhibited significant protein content, indicating that it is a valuable source of natural peptides with potent ACE inhibitory activity, which could be used for antihypertensive therapy. USE-S had the highest sugar and sulfate content, indicating its potential for use in energy production and as a source of fucoidan and alginate, which have various health benefits, including anticancer and anti-inflammatory effects. USE-R exhibited the highest antioxidant activities, which correlated with its high total phenolic and flavonoid content, underscoring a potential for combating oxidative stress-related diseases.

The identification of key compounds, such as sorbitol, glycerol, L-fucose, galactopyranose, palmitic acid, and various phenolic acids, along with their bioactivities, provides a basis for the development of functional foods, cosmetics, and pharmaceuticals. In addition, the significant correlations between chemical properties (TPC, TFC, TSC, RSC, and TPrC) and biological activities (ABTS, DPPH, FRAP, α-glucosidase inhibitory, and antihypertensive activity) suggest the potential of *U. pinnatifida* extracts in managing chronic conditions, such as diabetes and hypertension.

Understanding the molecular mechanisms and optimizing extraction methods will enhance the development of natural, safe, and effective health-promoting products from *U. pinnatifida*. Unlike previous studies, our research highlights the differential bioactive compound composition and bioactivities in the blade, sporophyll, and root of *U. pinnatifida*, offering more detailed and comprehensive insights. In summary, we not only confirmed the potential of *U. pinnatifida* as a source of natural bioactive compounds but also provided a foundation for its use in various industries, thereby contributing to the development of novel functional materials with significant health benefits. However, it is important to note the limitations of our current study, including the regional specificity of the seaweed samples, the use of a single extraction method, and the absence of certain advanced analytical techniques. Future research will focus on addressing these limitations by expanding regional sampling to include seaweed samples from diverse geographical locations, comparing the efficiency of various extraction methods, and incorporating advanced analytical techniques such as Ultra-Performance Liquid Chromatography-Electrospray Ionization-Quadrupole Time-of-Flight Mass Spectrometry/Mass Spectrometry, Nuclear Magnetic Resonance, Fourier Transform Infrared Spectroscopy, Raman Spectroscopy, X-ray Diffraction, Scanning Electron Microscopy, thermal analysis, etc. Additionally, we plan to isolate and characterize specific bioactive compounds and evaluate their efficacy and molecular mechanisms against specific diseases using cell and animal models.

## Figures and Tables

**Figure 1 marinedrugs-22-00344-f001:**
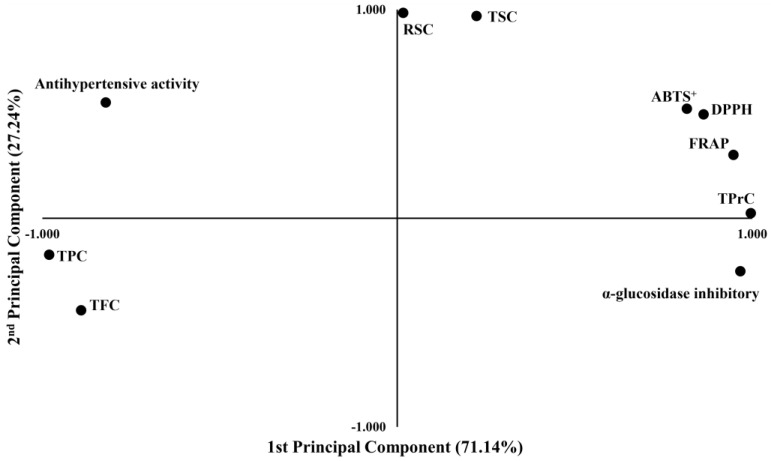
Principal component analysis of chemical properties and biological activity of the USE-s.

**Table 1 marinedrugs-22-00344-t001:** Proximate composition (%) of different parts of *U. pinnatifida*.

Parts	Moisture	Ash	Crude Lipid	Crude Protein	Carbohydrate
Blade	7.58 ± 0.01 ^b^	36.63 ± 0.16 ^b^	2.42 ± 0.62 ^c^	13.92 ± 0.88 ^a^	39.45 ± 0.42 ^b^
Sporophyll	8.28 ± 0.29 ^a^	20.31 ± 0.31 ^c^	6.34 ± 0.29 ^a^	13.74 ± 0.67 ^a^	51.33 ± 0.39 ^a^
Root	5.18 ± 0.27 ^c^	41.73 ± 0.05 ^a^	4.25 ± 0.31 ^b^	9.60 ± 0.79 ^b^	39.24 ± 0.36 ^b^

Values are expressed as the mean ± SD. Different letters indicate significant differences (*p* < 0.05) according to Tukey’s HSD multiple range test.

**Table 2 marinedrugs-22-00344-t002:** Extraction efficiency and physical parameters of extracts obtained from different parts of *U. pinnatifida*.

Parts	Extraction Efficiency (%)	Color					MRPs		
		L*	a*	b*	C*	H°	294 nm	420 nm	294/420
USE-B	80.40 ± 0.65 ^a^	32.30 ± 0.73 ^a^	13.32 ± 1.26 ^b^	13.65 ± 1.54 ^a^	19.07 ± 1.96 ^a^	45.65 ± 1.16 ^a^	2.580 ± 0.050 ^b^	0.188 ± 0.002 ^b^	13.703 ± 0.367 ^b^
USE-S	73.87 ± 0.32 ^b^	30.42 ± 0.42 ^b^	14.00 ± 0.45 ^a^	10.29 ± 0.36 ^b^	17.35 ± 0.46 ^b^	36.27 ± 1.30 ^b^	2.879 ± 0.068 ^a^	0.158 ± 0.004 ^c^	18.185 ± 0.279 ^a^
USE-R	65.93 ± 0.39 ^c^	27.68 ± 0.87 ^c^	11.56 ± 0.44 ^c^	6.96 ± 0.56 ^c^	13.28 ± 0.56 ^c^	30.67 ± 1.40 ^c^	2.886 ± 0.062 ^a^	0.225 ± 0.003 ^a^	12.844 ± 0.188 ^c^

Values are expressed as the mean ± SD. Different letters indicate significant differences (*p* < 0.05) according to Tukey’s HSD multiple range test.

**Table 3 marinedrugs-22-00344-t003:** Chemical properties of USE-s from different parts of *U. pinnatifida*.

Parts	Chemical Properties				
	Total Phenolic (mg PGE/g of Dry Sample)	Total Flavonoid (mg QE/g of Dry Sample)	Total Sugar (mg Glucose/g of Dry Sample)	Reducing Sugar (mg Glucose/g of Dry Sample)	Total Protein (mg BSA/g of Dry Sample)
USE-B	33.13 ± 0.14 ^b^	19.91 ± 0.54 ^b^	36.43 ± 0.75 ^c^	21.33 ± 0.51 ^c^	83.47 ± 1.76 ^a^
USE-S	30.11 ± 0.35 ^c^	9.22 ± 0.54 ^c^	97.35 ± 4.23 ^a^	56.44 ± 3.10 ^a^	84.93 ± 2.82 ^a^
USE-R	43.32 ± 0.19 ^a^	31.54 ± 1.63 ^a^	57.04 ± 1.39 ^b^	39.44 ± 3.61 ^b^	65.91 ± 3.53 ^b^

Values are expressed as the mean ± SD. Different letters indicate significant differences (*p* < 0.05) according to Tukey’s HSD multiple range test.

**Table 4 marinedrugs-22-00344-t004:** Monosaccharide compounds, sulfate content, and molecular weight analysis of USE-s from different parts of *U. pinnatifida*.

Parts	Monosaccharide Composition (%)					Sulfate Content (%)	Molecular Weight (Da)			
	Fucose	Galactose	Glucose	Xylose	Mannose		Peak No.	Mn	Mw	PI
USE-B	20.62 ± 1.07 ^b^	48.57 ± 0.27 ^a^	17.27 ± 0.73 ^a^	4.15 ± 0.13 ^c^	9.41 ± 0.19 ^c^	2.50 ± 0.10 ^b^	1	2263	2924	1.29
							2	609	664	1.09
							3	192	208	1.08
USE-S	41.99 ± 0.09 ^a^	25.04 ± 0.98 ^c^	14.20 ± 1.20 ^b^	8.41 ± 0.09 ^b^	10.37 ± 0.23 ^b^	7.76 ± 0.17 ^a^	1	2287	2914	1.27
							2	699	764	1.09
							3	199	214	1.08
USE-R	25.65 ± 2.25 ^c^	27.45 ± 0.05 ^b^	14.80 ± 0.30 ^b^	14.70 ± 2.60 ^a^	17.40 ± 0.60 ^a^	2.41 ± 0.20 ^c^	1	2307	3025	1.31
							2	606	663	1.09
							3	182	201	1.10

Values are expressed as the mean ± SD. Different letters indicate significant differences (*p* < 0.05) according to Tukey’s HSD multiple range test.

**Table 5 marinedrugs-22-00344-t005:** Identification of chemical compounds in USE-s by GC–MS.

Parts	Name	Classification	Sample (%)
1	Silanol, phosphate	Silane derivative	USE-S (0.75)
2	Glycerol	Sugar alcohol	USE-B (1.55), USE-S (1.08)
3	Butanedioic acid	Dicarboxylic acid	USE-B (1.70), USE-S (0.85), USE-R (2.85)
4	2-Butenedioic acid	Dicarboxylic acid	USE-S (0.35), USE-B (0.20)
5	Butanoic acid	Fatty acids	USE-B (0.20), USE-R (0.30)
6	DL-Pyroglutamic acid	Amino acid derivative	USE-B (2.15), USE-S (1.79)
7	L-Aspartic acid	α-Amino acid	USE-B (0.49)
8	N-heneicosane	Alkane	USE-B (0.50)
9	L-5-Oxoproline	α-Amino acid	USE-R (1.78)
10	Pentanedioic acid	Dicarboxylic acid	USE-B (0.71), USE-S (0.25), USE-R (1.09)
11	L-Fucose	Monosaccharide	USE-S (2.93), USE-R (0.48)
12	Galactopyranose	Monosaccharide	USE-S (0.47)
13	Citric acid	Tricarboxylic acid	USE-B (1.78), USE-S (1.93), USE-R (1.11)
14	Talose	Monosaccharide	USE-R (0.60)
15	Galactose	Monosaccharide	USE-S (0.67)
16	Sorbitol	Sugar alcohol	USE-B (53.57), USE-S (49.55), USE-R (33.82)
17	Palmitic Acid,	Fatty acids	USE-B (6.79), USE-S (4.25), USE-R (11.49)
18	Myo-Inositol	Sugar alcohol	USE-B (0.52), USE-S (1.79) USE-R (0.52)
19	Stearic acid	Fatty acid	USE-B (4.74), USE-S (2.80), USE-R (7.73)
20	Glyceryl-glycoside	Glycoside	USE-B (2.69), USE-S (0.27), USE-R (0.27)
21	9-Octadecenamide	Fatty amide	USE-S (1.46)
22	1-Monopalmitin	Monoacylglycerol	USE-S (2.40)
23	Glycerol monostearate	Monoacylglycerol	USE-S (2.75)

**Table 6 marinedrugs-22-00344-t006:** Biological activities of USE-s from different parts of *U. pinnatifida*.

Parts	Antioxidant Activities			Antidiabetic Activity	Antihypertensive Activity
	ABTS^+^	DPPH	FRAP	α-Glucosidase Inhibitory	
	IC_50_ Value (mg/mL)		EC_50_ Value (mg/mL)	IC_50_ Value (mg/mL)	IC_50_ Value (mg/mL)
USE-B	2.44 ± 0.23 ^b^	4.54 ± 0.14 ^b^	3.40 ± 0.17 ^b^	17.52 ± 0.71 ^a^	0.62 ± 0.01 ^c^
USE-S	3.70 ± 0.62 ^a^	5.96 ± 0.24 ^a^	4.02 ± 0.22 ^a^	14.69 ± 0.59 ^b^	0.76 ± 0.01 ^b^
USE-R	1.51 ± 0.12 ^c^	3.31 ± 0.24 ^c^	2.23 ± 0.13 ^c^	5.07 ± 0.45 ^c^	0.90 ± 0.02 ^a^
Standard	0.19 ± 0.01 ^d^	0.19 ± 0.01 ^d^	0.27 ± 0.01 ^d^	0.04 ± 0.01 ^d^	4.35 × 10^−5^ ± 0.01 ^d^

Values are expressed as the mean ± SD. Different letters indicate significant differences (*p* < 0.05) according to Tukey’s HSD multiple range test. The standard material for antioxidant activity is trolox, for antidiabetic activity is acarbose, and for antihypertensive activity, it is captopril.

**Table 7 marinedrugs-22-00344-t007:** Pearson’s correlation coefficients of the chemical properties and biological activities.

Trait	TPC	TFC	TSC	RSC	TPrC	ABTS^+^	DPPH	FRAP	α-Glucosidase Inhibitory	Antihypertensive Activity
TPC	1	0.961 **	−0.394 ^ns^	−0.200 ^ns^	−0.989 **	−0.925 **	−0.941 **	−0.992 **	−0.905 **	0.736 *
TFC		1	−0.632 ^ns^	−0.463 ^ns^	−0.909 **	−0.994 **	−0.998 **	−0.973 **	−0.753 **	0.521 *
TSC			1	0.975 **	0.513 ^ns^	0.628 *	0.576 *	0.825 **	−0.034 ^ns^	0.333 ^ns^
RSC				1	0.665 ^ns^	0.839 **	0.795 **	0.930 **	−0.235 ^ns^	0.516 *
TPrC					1	−0.974 **	−0.986 **	−0.926 **	0.959 **	−0.829 **
ABTS^+^						1	0.999 **	0.980 **	0.676 *	−0.423 ^ns^
DPPH							1	0.993 **	0.708 *	−0.464 ^ns^
FRAP								1	0.844 *	−0.644 ^ns^
α-glucosidase inhibitory									1	−0.954 **
Antihypertensive activity										1

* The correlation is significant at the 0.05 level. ** The correlation is significant at the 0.01 level. ns: not significant.

## Data Availability

The raw data supporting the conclusions of this article will be made available by the authors on request.

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
