# Peer review of "Subcritical Water Extraction of Undaria pinnatifida: Comparative Study of the Chemical Properties and Biological Activities across Different Parts"

_marinedrugs, 2024, doi:10.3390/md22080344_

Round 1
Reviewer 1 Report
Comments and Suggestions for Authors
The article presents a study on the subcritical water extraction of Undaria pinnatifida and the influence of the extraction method on the chemical properties and biological activity. The study is interesting and topical. Some corrections and clarifications are necessary.
Remarks
A. Introduction
1. The introduction should be completed with a more detailed description of the compounds found in algae, namely which types of polysaccharides, which phenolic compounds, which peptides, etc.
2. "Subcritical water extraction is a recently developed, eco-friendly, and clean extraction process that occurs at the critical temperature (374°C) and pressure (1–22 MPa) of water." - not true. Subcritical water extraction is done at temperatures (or pressures) lower than the critical ones. Besides, the temperature is the one that influences the polarity of the solvent.
B. Results and discussions
3. "These polysaccharides are known for their antitumor, antibacterial, immunostimulatory, and antiinflammatory effects" - requires bibliographic references.
4. The name phlorotannins is a bit pretentious. The total polyphenol content is determined by the Folin-Ciocalteu assay. For phlorotannins, the more specific method uses 2,4-dimethoxybenzaldehyde, a product that reacts specifically with 1,3-and 1,3,5-substituted phenols.
5. Table 5. The identification of the compounds by GC-MS was done by derivatization. For clarity, the presentation of the identified compounds should be done in free form, not as trimethylsilyl derivatives. Also, the presentation should be uniform (e.g. citric acid appears 2 times in the table - once for USE-B and then for USE-S, USE-R). How are Galactopyranose and D-(+)-Galactopyranose different in mass spectroscopy? Or beta-L-(-)-Fucopyranose, (R,S,R,R,S) and L-Fucose?
C. Materials and methods
6. Why were the algae thermally dried and not lyophilized, considering that some compounds with biological activity are thermolabile?
7. 3.4.2 – The abbreviations TPC and TFC should be defined (possibly in subtitle 3.4.2, for example Total phlorotannins content (TPC) ... ). The same for TSC and RSC.
8. 3.5.2. Antihypertensive activity
” Then, 20 μL of the sample solution were added to a 96-well microplate together with 20 μL of deionized water for the blank1 and blank2 wells. Next, a substrate buffer (20 μL) was added to the sample and the blank1 and blank2 wells, whereas blank2 wells contained 20 μL of deionized water. The enzyme solution was added to the sample and blank2 wells and incubated at 37 °C for 1 h. Subsequently, 200 μL of the indicator working solution were added and incubated at room temperature for 10 min. The microplate was read at 450 nm using a microplate reader and captopril as the reference." – is unclear. It should be reformulated. More details on the enzyme used are also needed. How do the test results compare to captopril?
9. 3.5.3.α-. glucosidase inhibitory activity
-What is pNPG?
D. Conclusions
10. What is new in this study compared to paper [10] (Park, J.-S.; Han, J.-M.; Shin, Y.-N.; Park, Y.-S.; Shin, Y.-R. ; Park, S.-W.; Lee, H.-J., Exploring bioactive compounds in brown seaweeds 2023 , (6), 328)? The novelties should be highlighted in the Conclusions and in the abstract.
Author Response
Response to the comments of the Reviewer 1
We greatly appreciate to the Reviewer for evaluating our manuscript. On the following text, we respond to the Reviewer’s comments point by point. The changes in the revised manuscript have been marked in red color. All line numbers mentioned in the responses to comments refer to the numbers that appear on the right margin of the text of the revised manuscript.
General Comment: The article presents a study on the subcritical water extraction of Undaria pinnatifida and the influence of the extraction method on the chemical properties and biological activity. The study is interesting and topical. Some corrections and clarifications are necessary.
Response: Thank you very much for your valuable times and efforts to review our manuscript.
Comment 1. Introduction: The introduction should be completed with a more detailed description of the compounds found in algae, namely which types of polysaccharides, which phenolic compounds, which peptides, etc.
Response: Thank you for your valuable feedback and insightful comments. Based on your suggestions, we have revised the introduction section to include a more detailed description of the compounds found in brown algae, specifically focusing on polysaccharides, phenolic compounds, and other bioactive substances. (Lines 58-70).
Comment 2. Introduction: "Subcritical water extraction is a recently developed, eco-friendly, and clean extraction process that occurs at the critical temperature (374°C) and pressure (1–22 MPa) of water." - not true. Subcritical water extraction is done at temperatures (or pressures) lower than the critical ones. Besides, the temperature is the one that influences the polarity of the solvent.
Response: We apologize for the mistake. We have corrected the sentence as “Subcritical water extraction is an emerging, eco-friendly, and clean extraction process that occurs at the temperature ranging 100-374°C and pressure (1–22 MPa) of water” (Lines 91-92).
Comment 3. B. Results and discussions: "These polysaccharides are known for their antitumor, antibacterial, immunostimulatory, and antiinflammatory effects" - requires bibliographic references.
Response: Thank you for your valuable feedback and insightful comments. The reference is added in the revised manuscript (Line 146).
Comment 4. B. Results and discussions: The name phlorotannins is a bit pretentious. The total polyphenol content is determined by the Folin-Ciocalteu assay. For phlorotannins, the more specific method uses 2,4-dimethoxybenzaldehyde, a product that reacts specifically with 1,3-and 1,3,5-substituted phenols.
Response: Thank you for your valuable comment and suggestion. We appreciate your input on this matter. In our study, we used the Folin-Ciocalteu assay to measure the total phenolic content, and we acknowledge that this method is non-specific and measures the overall polyphenol content rather than specifically identifying phlorotannins. We have revised the manuscript to refer to the measured content as "Total Phenolic Content" instead of "phlorotannins" to reflect this accurately.
We are grateful for your suggestion to use the 2,4-dimethoxybenzaldehyde method to measure phlorotannins specifically. Although we did not employ this method in the current study, we recognize its importance and specificity for phlorotannins. In future research, we will consider its application to provide more detailed and specific data.
Comment 5. B. Results and discussions: Table 5. The identification of the compounds by GC-MS was done by derivatization. For clarity, the presentation of the identified compounds should be done in free form, not as trimethylsilyl derivatives. Also, the presentation should be uniform (e.g. citric acid appears 2 times in the table - once for USE-B and then for USE-S, USE-R). How are Galactopyranose and D-(+)-Galactopyranose different in mass spectroscopy? Or beta-L-(-)-Fucopyranose, (R,S,R,R,S) and L-Fucose?
Response: Thank you for your valuable comments and suggestions. We have revised Table 5 to present the identified compounds in their free form rather than as trimethylsilyl derivatives, ensuring they are clearly represented in their natural states. We have also ensured that each compound is listed only once in the table, even if it was identified in multiple samples; for instance, citric acid is now listed once with indications of the samples (USE-B, USE-S, USE-R) in which it was found. Regarding the nomenclature, Galactopyranose and D-(+)-Galactopyranose, as well as beta-L-(-)-Fucopyranose and L-Fucose, refer to the same chemical entities (D-galactose and L-fucose, respectively) and do not imply different compounds in the context of mass spectrometry. Thus, these names represent the same molecules.
Comment 6. C. Materials and methods Why were the algae thermally dried and not lyophilized, considering that some compounds with biological activity are thermolabile?
Response: Thank you for your valuable comment. Our research aimed to implement cost-effective, eco-friendly, and energy-saving procedures. Thermal drying was chosen to achieve these goals, as lyophilization is an expensive process that does not align well with our economic and environmental sustainability objectives.
While freeze-drying is more suitable for preserving thermolabile pigments and compounds, thermal drying is adequate for substances that are not highly sensitive to heat and provides a more economical solution. Additionally, some studies have shown that thermal drying can be superior to freeze-drying in certain aspects, such as higher reducing power, free radical scavenging, and metal chelating activities.
“Que, F., Mao, L., Fang, X., & Wu, T. (2008). Comparison of hot air‐drying and freeze‐drying on the physicochemical properties and antioxidant activities of pumpkin (Cucurbita moschata Duch.) flours. International journal of food science & technology, 43(7), 1195-1201.”
Moreover, the samples we received were by-products already subjected to initial sun drying near the sea after harvesting. Therefore, we performed minimal additional thermal drying. Lastly, during subcritical water hydrolysis, the samples were exposed to high temperatures (180°C), which means any potential thermal degradation would have already occurred at this stage.
In future research, we plan to collect untreated samples and compare the effects of various pretreatment methods on compound preservation, such as lyophilization, thermal drying, and sun drying. Your feedback will significantly enhance the quality of our future studies.
Comment 7. C. Materials and methods 3.4.2 – The abbreviations TPC and TFC should be defined (possibly in subtitle 3.4.2, for example Total phlorotannins content (TPC) ... ). The same for TSC and RSC.
Response: Thank you for your valuable comment. We have revised the manuscript to include the definitions of the abbreviations TPC, TFC, TSC, and RSC etc. at their first occurrence in the text. Additionally, we have ensured that these abbreviations are clearly defined in the relevant sections to enhance clarity.
Comment 8. C. Materials and methods 3.5.2. Antihypertensive activity
” Then, 20 μL of the sample solution were added to a 96-well microplate together with 20 μL of deionized water for the blank1 and blank2 wells. Next, a substrate buffer (20 μL) was added to the sample and the blank1 and blank2 wells, whereas blank2 wells contained 20 μL of deionized water. The enzyme solution was added to the sample and blank2 wells and incubated at 37 °C for 1 h. Subsequently, 200 μL of the indicator working solution were added and incubated at room temperature for 10 min. The microplate was read at 450 nm using a microplate reader and captopril as the reference." – is unclear. It should be reformulated. More details on the enzyme used are also needed. How do the test results compare to captopril?
Response: Thank you for your comment regarding the unclear description of the method. We appreciate your input to ensure our explanations are clear.
First, we used the ACE Kit purchased from Dojindo Molecular Technologies to measure antihypertensive activity. Based on the provided manual, we have revised the methods section to provide a clearer explanation.
Although the exact origin and manufacturer of the enzyme used in the ACE Kit are not specified, we will explain the primary mechanism of antihypertensive activity to aid in understanding.

Figure. Principle of the ACE inhibition assay using ACE Kit – WST
[Dojindo Molecular Technologies. ACE Kit - WST A502 Product Manual. Available from: Dojindo Molecular Technologies; https://www.dojindo.com/manual/A502/ [20 July, 2024].]
[Angiotensin-converting enzyme (ACE) is one of the key elements responsible for vasopressor action. ACE converts angiotensin-I to angiotensin-ll, a potent vasopressor, in the rennin-angiotensin system and contributes to increasing blood pressure by inactivating bradykinin, a strong antihypertensive peptide. Recently, various functional foods have received attention because of their inhibitory activity toward ACE.
ACE activity is conventionally determined by UV measurement of the hippuric acid produced from the synthetic substrate Hyppuryl-His-Leu. However, the assay process is complicated and requires organic solvent. In this kit, a safe and straightforward modified method has been developed.
The colorimetric detection system in the ACE Kit-WST determines the amount of 3-hydroxybutyric acid (3HB) generated from 3-hydroxybutyryl-Gly-Gly-Gly by ACE. The kit is designed for 96-well microplate assays and is suitable for multiple sample measurements. No organic solvent extraction is required. The ACE Kit - WST assay is safe, simple, and provides highly reproducible data.]
Captopril was used as a standard, not to quantify the antihypertensive activity of the samples but to compare their IC50 values with those of a known antihypertensive drug. The IC50 value of captopril, measured in the same manner as other samples, is provided in Table 6 and Line 447. Thank you again for your valuable feedback.
Comment 9. C. Materials and methods 3.5.3.α-. glucosidase inhibitory activity
-What is pNPG?
Response: Thank you for your comment. pNPG stands for p-nitrophenyl-α-D-glucopyranoside. It is a synthetic substrate commonly used in α-glucosidase inhibition assays. When pNPG is hydrolyzed by α-glucosidase, it releases p-nitrophenol, which can be quantified by measuring the absorbance at 405 nm. This allows us to determine the inhibitory activity of the tested samples against α-glucosidase.
Comment 10. D. Conclusions What is new in this study compared to paper [10] (Park, J.-S.; Han, J.-M.; Shin, Y.-N.; Park, Y.-S.; Shin, Y.-R.; Park, S.-W.; Lee, H.-J., Exploring bioactive compounds in brown seaweeds 2023, (6), 328)? The novelties should be highlighted in the Conclusions and in the abstract.
Response: Thank you for your comment. The current study builds on the work of Park et al. (2023) by introducing several key innovations and advancements. While the previous study utilized subcritical water extraction to extract and characterize bioactive compounds from various brown seaweeds, our research extends this by focusing specifically on Undaria pinnatifida. We investigate the variation in compound composition and bioactivity across different parts of the seaweed (blade, sporophyll, and root), providing a more detailed analysis.

Reviewer 2 Report
Comments and Suggestions for Authors
My comments are in the attached file.

Minor revision
Author Response
Response to the comments of the Reviewer 2
We greatly appreciate to the Reviewer for evaluating our manuscript. On the following text, we respond to the Reviewer’s comments point by point. The changes in the revised manuscript have been marked in red color. All line numbers mentioned in the responses to comments refer to the numbers that appear on the right margin of the text of the revised manuscript.
Comment 1: Abstract; Line 20: Why this condition? Why not vary time, temperature and pressure matrix?
Response: Thank you for your insightful comment. We selected the conditions of 180 °C, 3 MPa, and 30 min based on extensive preliminary studies. Our research showed that temperatures around 180-210 °C yielded the highest bioactive compound content. We chose 180°C as it balanced efficiency and energy consumption while avoiding compound degradation.
“Kim, S. Y., Roy, V. C., Park, J. S., & Chun, B. S. (2024). Extraction and characterization of bioactive compounds from brown seaweed (Undaria pinnatifida) sporophyll using two sequential green extraction techniques. Algal Research, 77, 103330.”
“Park, J. S., Jeong, Y. R., & Chun, B. S. (2019). Physiological activities and bioactive compound from laver (Pyropia yezoensis) hydrolysates by using subcritical water hydrolysis. The Journal of Supercritical Fluids, 148, 130-136.”
“Park, J. S., Han, J. M., Surendhiran, D., & Chun, B. S. (2022). Physicochemical and biofunctional properties of Sargassum thunbergii extracts obtained from subcritical water extraction and conventional solvent extraction. The Journal of Supercritical Fluids, 182, 105535.”
The pressure was set at 3 MPa to maintain water in its subcritical state, as the literature indicates that pressure has no significant effect on compound recovery efficiency.
“Zhang, J., Wen, C., Zhang, H., Duan, Y., & Ma, H. (2020). Recent advances in the extraction of bioactive compounds with subcritical water: A review. Trends in Food Science & Technology, 95, 183-195.”
We fixed the extraction time at 30 min based on empirical knowledge. Shorter times may not sufficiently dissolve functional compounds like phenolic compounds from seaweed, while longer times can lead to compound degradation or unwanted reactions.
“Yan, L., Cao, Y., & Zheng, G. (2017). Optimization of subcritical water extraction of phenolic antioxidants from pomegranate (Punica granatum L.) peel by response surface methodology. Analytical Methods, 9(32), 4647-4656.”
Comment 2: Abstract; Line 30: So what extraction technology was used as a control?
Response: Thank you for your comment. In this study, we did not use other extraction methods (such as solvent extraction) as controls. The aim of our research was to compare the differences in chemical properties and biological activities of different parts of Undaria pinnatifida using subcritical water extraction. Our previous studies compared subcritical water extraction with conventional solvent extraction methods (ethanol, methanol, hot water, etc.). We demonstrated that subcritical water extraction is superior in terms of chemical properties and bioactivity. For detailed comparisons, please refer to the following references:
“Park, J. S., Han, J. M., Surendhiran, D., & Chun, B. S. (2022). Physicochemical and biofunctional properties of Sargassum thunbergii extracts obtained from subcritical water extraction and conventional solvent extraction. The Journal of Supercritical Fluids, 182, 105535.”
“Jeong, Y. R., Park, J. S., Nkurunziza, D., Cho, Y. J., & Chun, B. S. (2021). Valorization of blue mussel for the recovery of free amino acids rich products by subcritical water hydrolysis. The Journal of Supercritical Fluids, 169, 105135.”
“Ho, T. C., & Chun, B. S. (2019). Extraction of bioactive compounds from Pseuderanthemum palatiferum (Nees) Radlk. using subcritical water and conventional solvents: A Comparison Study. Journal of food science, 84(5), 1201-1207.”
Comment 3: Abstract; Line 30: Also perform a principal component analysis.
Response: We sincerely appreciate your valuable feedback. Following your suggestion, we have conducted a new statistical analysis, including principal component analysis (PCA), and incorporated the related discussion in the revised manuscript (Lines 464-487). The PCA has provided additional insights into the data, enhancing the overall understanding of the results.
Comment 4: Keywords; Line 34-35; Provide list of abbreviations.
Response: Thank you for your suggestion. We have added a list of abbreviations following the abstract in the revised manuscript. (Lines 39-45)
Comment 5: Introduction; Line 43; As of which year?
Response: Thank you for your comment. The year of production has been added to the revised manuscript. (Line 54).
Comment 6: Introduction; Line 69; SWE is not recent. It is an emerging technology. The phrase “emerging technology” refers to a technology that is either (1) in the development phase with a high possibility of commercialization in the next five years or (2) currently commercialized, although it only accounts for a small fraction of the market for applications in the food industry.
Reference
Hernández-Hernández, H. M., Moreno-Vilet, L., & Villanueva-Rodríguez, S. J. (2019). Current status of emerging food processing technologies in Latin America: Novel non-thermal processing. Innovative Food Science & Emerging Technologies, 58, 102233.
Boateng, I. D. (2023). Recent advances in combined Avant-garde technologies (thermal-thermal, non-thermal-non-thermal, and thermal-non-thermal matrix) to extract polyphenols from agro byproducts. Journal of Food and Drug Analysis, 31(4), 552.
Zhang, J., Wen, C., Zhang, H., Duan, Y., & Ma, H. (2020). Recent advances in the extraction of bioactive compounds with subcritical water: A review. Trends in Food Science & Technology, 95, 183-195.
Response: Thank you for your insightful comment and the provided references. We appreciate the clarification regarding the definition of "emerging technology." In the revised manuscript, we have updated the terminology to reflect that SWE is indeed an emerging technology. We have also incorporated the provided references to support this characterization. (Lines 91-92)
Comment 7: Results and Discussion; Line 113: Statistical analysis have to be discussed comprehensively in the results and discussion section as well. Again, the authors spend too much space in general information about irrelevant topics. I have read the results and discussion section carefully and have not found a sign that the appropriate discussion of the results was obtained. The authors must focus on discussing and comparing their findings with previous reports on this field. The current form of results and discussion is more like general reporting. Authors spend too much space on general information. Besides, the authors should talk about the chemistry behind your results. talk about the chemistry/ science behind your results. A comprehensive discussion should include the results, the trend(s), the reasons for the trend(s) obtained, and a comparison with other studies. What is the chemistry and science behind your results?
Response: Thank you for your detailed feedback. We understand the importance of providing a comprehensive discussion of our results. Based on your comments, we have made significant revisions to the Results and Discussion section to address the following points. As you suggested, we have included a comprehensive discussion of the statistical analyses, including ANOVA results, post hoc analysis, Pearson correlation, and PCA. Additionally, we have provided an in-depth discussion of the results and trends observed in our data, referencing various previous studies to support our findings. We have also explained the underlying chemical and scientific mechanisms to understand the results better. These revisions are throughout the revised manuscript, and all changes have been highlighted in red for your convenience.
Comment 8: Materials and Methods; Line 379; Provide latitude and longitude.
Response: Thank you for your comment. The Undaria pinnatifida used in this study was obtained from a company based at 18 Dangsa-ro 2-gil, Gijang-eup, Gijang-gun, Busan, Republic of Korea. The seaweed was grown in nearby coastal areas. The approximate latitude and longitude of the location are 35.187753° N, 129.211470° E. We have included this information in the revised manuscript. (Line 491)
Comment 9: Materials and Methods; Line 414; This is the same as section 3.2.
Response: Thank you for pointing out this redundancy. We apologize for the oversight and have removed the duplicate section in the revised manuscript.
Comment 10: Materials and Methods; Line 423: Also calculate the chroma and hue angle.
Response: Thank you for your valuable comment. We have calculated the chroma (C*) and hue angle (H°) in addition to the L*, a*, and b* values. These calculations have been included in the revised manuscript, and the discussion section has been updated accordingly to reflect the significance of these colorimetric parameters. (Lines 182-196, 536)
Comment 11: Line 435: What was the concentration? The materials used and the details and conditions of experimental procedures have to be described with sufficient clarity, thus allowing qualified operators to repeat the work.
Response: Thank you for your valuable comment. We appreciate your attention to detail. The revised manuscript now includes detailed information on the preparation and dilution of the samples (Lines 517-521), ensuring that the experimental procedures are described with sufficient clarity for reproducibility.
Comment 12: Line 443: Each analyte has different extraction process. So what was the extraction process before analysis. The same applies with other analyte of interest.
Response: Thank you for your comment. We did not use separate extraction processes for each analyte before analysis. Instead, the freeze-dried extract powder was dissolved in distilled water at appropriate concentrations for each assay. This approach was applied uniformly across all analytes of interest.
Comment 13: Line 471: Provide the hplc chromatogram in the supplementary figure.
Response: Thank you for your valuable suggestion. We have provided the chromatograms for each analysis in the supplementary file.
Comment 14: Line 495: Provide the hplc chromatogram in the supplementary figure.
Response: Thank you for your valuable suggestion. We have provided the chromatograms for each analysis in the supplementary file.
Comment 15: Line 514: How did you ensure that there were no false positives?
Response: Thank you for your insightful comment. We did not specifically analyze for false positives in this study. However, we ensured the accuracy of our experiments by using appropriate controls and standards. In the antioxidant assays, we used Trolox as a standard; in the antihypertensive assays, we used Captopril; and in the α-glucosidase inhibitory assays, we used Acarbose. Additionally, all experiments were conducted in triplicate at different concentrations to ensure consistent results. These measures helped us to secure the reliability of our findings.
Comment 16: Line 563: Also perform principal component analysis.
Response: Thank you for your suggestion. As mentioned in response to Comment 3, we have conducted a new statistical analysis, including principal component analysis (PCA), and incorporated the related discussion in the revised manuscript (Lines 464-487). The PCA has provided additional insights into the data, enhancing the overall understanding of the results.
Comment 17: Line 588: Also, talk about this study's limitations and the future studies required.
Response: Thank you for your valuable feedback. Our study has limitations, such as the regional specificity of seaweed samples, the use of a single extraction method, and the lack of advanced analytical techniques. Future research will address these by expanding geographical sampling, comparing different extraction methods, and incorporating techniques like UPLC-ESI-QTOF-MS/MS, NMR, FT-IR, Raman, XRD, SEM, and thermal analysis. Additionally, we aim to isolate and characterize bioactive compounds and assess their efficacy in disease models. We have incorporated this discussion into the conclusion section of the revised manuscript. (Lines 699-711)
Comment 18: Line 589: The manuscript article requires revision in grammar, sentence structure, and reference format. Overused Passive voice in the manuscript seems hard to read. Please carefully check the sections: introduction, results, discussion, and conclusions. Please try to reword the phrases in the active voice. Grammar and punctuation mistakes. For consistency, please use the manuscript in just one English style (a non-variant British or British style, American style, etc.). There are phrases with the verb in the wrong tense. Sentences with words misspelled. Words are overused or unnecessary. Nouns without determiner or unnecessary
Response: Thank you for your detailed feedback regarding the language and style of the manuscript. We appreciate the importance of clear and consistent language. Due to the short revision time provided by the journal, we were unable to address all the language corrections in the current version of the manuscript. However, we have informed a professional language editing service of your comments and are in the process of having the manuscript thoroughly revised. We assure you that the final version will address the issues you raised and meet the high standards expected.

Round 2
Reviewer 1 Report
Comments and Suggestions for Authors
The authors answered and clarified all the problems highlighted in the first review. The quality of the article has improved.
Reviewer 2 Report
Comments and Suggestions for Authors
The manuscript can be accepted for publication in it’s current form
Comments on the Quality of English LanguageMinor